# Inferring Cell–Cell Communications from Spatially Resolved Transcriptomics Data Using a Bayesian Tweedie Model

**DOI:** 10.3390/genes14071368

**Published:** 2023-06-28

**Authors:** Dongyuan Wu, Jeremy T. Gaskins, Michael Sekula, Susmita Datta

**Affiliations:** 1Department of Biostatistics, University of Florida, Gainesville, FL 32603, USA; dongyuanwu@ufl.edu; 2Department of Bioinformatics and Biostatistics, University of Louisville, Louisville, KY 40202, USA; jeremy.gaskins@louisville.edu (J.T.G.); michael.sekula@louisville.edu (M.S.)

**Keywords:** cellular communication, spatial transcriptomics, generalized linear regression model, Bayesian modeling, Tweedie distribution

## Abstract

Cellular communication through biochemical signaling is fundamental to every biological activity. Investigating cell signaling diffusions across cell types can further help understand biological mechanisms. In recent years, this has become an important research topic as single-cell sequencing technologies have matured. However, cell signaling activities are spatially constrained, and single-cell data cannot provide spatial information for each cell. This issue may cause a high false discovery rate, and using spatially resolved transcriptomics data is necessary. On the other hand, as far as we know, most existing methods focus on providing an ad hoc measurement to estimate intercellular communication instead of relying on a statistical model. It is undeniable that descriptive statistics are straightforward and accessible, but a suitable statistical model can provide more accurate and reliable inference. In this way, we propose a generalized linear regression model to infer cellular communications from spatially resolved transcriptomics data, especially spot-based data. Our BAyesian Tweedie modeling of COMmunications (BATCOM) method estimates the communication scores between cell types with the consideration of their corresponding distances. Due to the properties of the regression model, BATCOM naturally provides the direction of the communication between cell types and the interaction of ligands and receptors that other approaches cannot offer. We conduct simulation studies to assess the performance under different scenarios. We also employ BATCOM in a real-data application and compare it with other existing algorithms. In summary, our innovative model can fill gaps in the inference of cell–cell communication and provide a robust and straightforward result.

## 1. Introduction

Different biochemical signalings from cellular communications control different activities of living organisms, which highlights the importance of understanding cell–cell communications (CCC) on biological processes and mechanisms [1,2]. In practice, we infer the CCC from some known ligand–receptor (LR) pairs because the interaction of LRs mediates communication. As single-cell RNA sequencing (scRNA-seq) technologies have matured, researchers have gradually gained opportunities to investigate CCC from scRNA-seq data since we know more about the ligand and receptor gene expression information and cell type annotation at the cellular level. Several approaches have been proposed for inferring the CCC from scRNA-seq data. For example, CellPhoneDB [3] calculates the mean of the average ligand expression level for one cell type and the average receptor expression level for another cell type and conducts a permutation test to determine the significance of this LR pair between two cell types. Instead of using the mean or product to measure the communication between two cell types, SingleCellSingleR [4] introduces a regularized product score for an LR pair and provides an ad hoc benchmark to decide an appropriate score threshold. In addition, CellChat [5] offers a more complicated measurement to reflect the interaction strength of an LR pair between cell types, but it also utilizes the permutation test to identify statistical significance. Even though these methods can infer cellular communications to some extent, one of the common limitations is that they do not consider the spatial information for each cell, which is crucial for cell signaling activities but lost in single-cell data. This restriction may lead to high false discovery rates in discovering intercellular communications [1].

Fortunately, in recent years, the development of various spatially resolved transcriptomics (SRT) technologies has made it possible to access cellular locations, opening up new opportunities to incorporate physical distances of cells into CCC analysis. Giotto [6] provides a similar communication measurement as CellPhoneDB [3], but it incorporates the spatial information. Its computations focus on proximal cells, and its permutation tests shuffle cell locations within the same cell type rather than mixing the cell type annotations. SpaOTsc [7] treats CCC analysis as an optimal transport problem, using a random forest model to estimate the spatial distance of a signaling pathway and adjusting the cost matrix of the optimal transport plan by incorporating inferred spatial constraints. COMMOT [8] shares a similar framework with SpaOTsc [7] but accounts for the competition between cells in the signaling analysis. SpaTalk [9] evaluates the communication scores using intercellular and intracellular scores. The intercellular score is based on the number of one-hop neighbor nodes of receiver cell types for each sender cell type, while the intracellular score is computed from their integrated LR transcription factor knowledge graph. More comprehensive introductions to the existing CCC methods can be found in [1,10].

One should note that the existing approaches for SRT data also have their own drawbacks. The existing spatial CCC approaches only consider the single-cell resolution data from technologies such as seqFISH+ [11] and STARmap [12]. However, some spot-based technologies, such as the widely used 10X Visium [13,14] and Slide-seqV2 [15], detect gene expression levels based on spots, which means that each pixel location may contain several cells. This challenge persists even with high-resolution technologies that can reach the size of mammalian cells, as cells may overlap with each other [16]. Thus, it is valuable to interpret the mixture of multiple cell types and their corresponding proportions for CCC analysis from the spot-based data. Some recent methods using cell type deconvolution, such as RCTD [16], SPOTlight [17], and STRIDE [18], have considered the mixture issue of the spot-based data, but few CCC approaches have been proposed to address it. In addition, most of the existing CCC methods focus on providing an ad hoc measurement to estimate intercellular communication instead of relying on a statistical model. While it is undeniable that descriptive statistics are straightforward and accessible to interpret, a suitable statistical model is needed to provide more accurate and reliable estimation and inference.

In this paper, we introduce a novel generalized linear regression model with compound Poisson–Gamma distributions, also known as Tweedie distribution with p∈(1,2), to infer the communications between cell types. The model combines the physical locations of spots/cells and the proportions of cell types to estimate the signaling strength from one cell type to another. Its unique structure allows it to handle both spot-based SRT data and SRT data at the single-cell resolution, and it is able to consider the communication between different cell types simultaneously for a particular LR pair. For spot-based SRT data, our model uses a convolution strategy to integrate the possible interactions between the cell types at the sender spots and the cell types at the receiver spots to the average spot-to-spot communication scores. Furthermore, due to the properties of the regression model, our proposed method naturally provides the direction of the association between cell type communication and LR interaction that other approaches cannot offer. Since we utilize Bayesian inference for this model, we refer to our approach as BATCOM, which is shorthand for BAyesian Tweedie modeling of COMmunications.

The rest of this manuscript is organized as follows. In Section 2, we define the model structure and demonstrate how our model estimates the communication strength from one cell type to another based on the gene expression matrix and the spot locations from SRT data. In Section 3, both simulation studies and case studies are conducted to display the usability, reliability, and robustness of the proposed model. Finally, we summarize our conclusions and give a discussion in Section 4.

## 2. Materials and Methods

The workflow of our proposed approach is summarized in Figure 1. Firstly, we require the gene expression matrix from the SRT data with a list of known LR pairs to calculate the communication scores. We also need the spatial information (i.e., coordinate locations) of spots/cells to determine the distances between each pair of spots/cells. Next, we combine cell type annotations for spots/cells obtained from some upstream cell type deconvolution tools, such as RCTD [16], STRIDE [18], and SPOTlight [17]. Specifically, for spot-based SRT data, we should have information on the proportion of each cell type within each spot. On the other hand, for SRT data at the single-cell resolution, it is also easy to generate a matrix that identifies the cell type for each cell, as these data are a special case for our approach where the membership proportion is 1 for the corresponding cell type and 0 otherwise. For clarity, we will focus exclusively on spot-based SRT data in the following sections. After fitting a generalized linear regression model, the communication strength between different cell types is obtained from the regression coefficients. One can also display the communication strengths between cell types in detail using a heatmap and construct a network to visualize cell type communication.

### 2.1. Spot-to-Spot Communication Score

Since the spot-based SRT data only provide the gene expression level for each spot, we define the communication score $C_{i,j}^{k}$ (i,j=1,2,…,N) from sender spot *i* to receiver spot *j* for the LR pair *k* as
(1)$$C_{i,j}^{k}=L_{i}^{k}\times R_{j}^{k},$$
where $L_{i}^{k}$ and $R_{j}^{k}$ are the expression level of ligand *L* at spot *i* and the expression level of receptor *R* at spot *j*, separately. Specifically, $L_{i}^{k}$ and $R_{j}^{k}$ can be estimated using the arithmetic mean if the ligand *L* or receptor *R* consists of more than one subunit. In other words,
$$L_{i}^{k}=\frac{1}{s_{L}}\sum_{s=1}^{s_{L}}y_{i}^{L_{k,s}},$$
$$R_{j}^{k}=\frac{1}{s_{R}}\sum_{s=1}^{s_{R}}y_{j}^{R_{k,s}},$$
where $s_{L}$ and $s_{R}$ denote the numbers of subunits of ligand *L* and receptor *R*, and $y_{i}$ and $y_{j}$ are the corresponding normalized gene expression levels of spot *i* and spot *j* in SRT data. Although the geometric mean of subunits can be an alternative option for estimation, this will introduce a significantly larger number of zeros present in $C_{i,j}^{k}$ compared to the arithmetic mean.

While public databases such as CellChatDB [5], CellPhoneDB [3], and CellTalkDB [19] offer an extensive list of LR pairs, it is crucial to note that not all pairs are relevant for the analysis since they may not exist in the data being studied. Therefore, we remove LR pairs whose spot-to-spot communication scores $C_{i,j}$ contain more than 98% zeros. Once we obtain the communication scores $C_{i,j}^{k}$ for each relevant LR pair *k*, we can calculate spot-to-spot communication scores for a specific signaling pathway or the entire system by summing up the corresponding LR pairs.

The $C_{i,j}^{k}$ communication scores in Equation (Equation 1) will be the outcome variables in the regression modeling yielding $N^{2}$ observations. In practice, this dimensionality can be reduced by only considering (i,j) pairs that are within a certain distance of each other.

### 2.2. BATCOM Model Structure

To model the signaling strength for LR pair *k* across all combinations of cell types from the communication scores between two spots, we propose a generalized linear regression model
(2)$$\begin{array}{c} g\left(E(C_{i,j}^{k})\right)=\beta_{0}^{k}+\left[\sum_{g_{1},g_{2}}\beta_{g_{1},g_{2}}^{k}M_{i,g_{1}}M_{j,g_{2}}\right]\times \exp\left(-\rho D_{i,j}\right)+\nu_{i}^{L}+\nu_{j}^{R},\;i,j=1,2,\ldots ,N, \end{array}$$
where $M_{i,g_{1}}$ and $M_{j,g_{2}}$ are the proportions of cell types $g_{1}\left(g_{1}=1,2,\ldots ,G\right)$ at the sender spot *i* and $g_{2}\left(g_{2}=1,2,\ldots ,G\right)$ at the receiver spot *j*. Although the model typically includes $G^{2}$ interaction terms to account for the communications among *G* cell types, some of these product terms may be filtered out in practice due to minimal or non-existent observations, or they may be based on prior knowledge.

In addition, ρ>0 is a communication constraint parameter, and $D_{i,j}$ is a suitably chosen distance metric between spots *i* and *j*, which we considered to be Euclidean in our applications. As spots *i* and *j* are spatially further apart ($D_{i,j}$ increasing), they have less ability to communicate. This effect is captured by the $\exp(-\rho D_{i,j})$ term in Equation (Equation 2), which down-weights the impact of the cell type memberships as distance increases. Larger values of ρ represent a faster spatial decay such that only adjacent spots may communicate, while smaller values of ρ allow communication across longer distances. However, the specific value of ρ should be chosen based on the scale of distances in the dataset. In this paper, we scaled $D_{ij}$ so that adjacent spots had a distance of 1.

Based on Equation (Equation 1), $C_{i,j}^{k}\;\left(i,j=1,2,\ldots ,N\right)$ are not independent of each other as they partially come from the same spot. For example, when i=1, all $C_{1,j}^{k}\;\left(j=1,2,\ldots ,N\right)$ should be correlated with each other because they all depend on the same sender spot expression $L_{1}^{k}$. Thus, Equation (Equation 2) includes two random effect parameters $\nu_{i}^{L}$ and $\nu_{j}^{R}$ to introduce correlation around the corresponding sender spot *i* for ligand *L* and receiver spot *j* for receptor *R*, respectively. Returning to our example, if spot 1 exhibits high expression levels for a specific ligand $L_{1}^{k}$, it will result in the corresponding communication scores $C_{1,j}^{k}$ being large or above average for all *j*. Neglecting to account for the shared structure across $C_{1,j}^{k}$ for j=1,2,…,N may lead to an overestimation of the effect of the cell types most prevalent in spot 1. To address this issue, we introduce the inclusion of a large $\nu_{1}^{L}$ to capture the characteristics of spot 1, accounting for its high expression of ligand *L* and communication scores $C_{1,j}$. By incorporating this additional variability, the remaining variation in $C_{1,j}^{k}$ becomes associated with the primary target of interest: the cell type combinations.

The regression coefficients $\beta_{g_{1},g_{2}}^{k}$ in Equation (Equation 2) are the parameters of interest in our model and thus reflect the communication strength of LR pair *k* from sender cell type $g_{1}$ to receiver cell type $g_{2}$. A positive $\beta_{g_{1},g_{2}}^{k}$ indicates that as the memberships of cell type $g_{1}$ at the sender spot and cell type $g_{2}$ at the receiver spot jointly increase, the communication is predicted to increase. Conversely, negative coefficients suggest that larger cell type memberships will decrease the spot-to-spot communication of LR pair *k*. In this way, our approach is capable of deconvoluting the mean spot-to-spot communication scores into the interactions between the cell types at the sender spot and the cell types at the receiver spot.

It is important to note that the original gene expression matrix of SRT data is usually sparse. Moreover, if the ligand *L* or the receptor *R* is not expressed, the communication score $C_{i,j}^{k}$ will be zero according to Equation (Equation 1). As a result, $C_{i,j}^{k}\left(i,j=1,2,\ldots ,N\right)$ is expected to be a sparse vector with continuous positive scores in the non-zero positions. Considering this property of the data, one common choice would be fitting a zero-inflated or hurdle model. However, both models require two sets of coefficients to account for the probability of zeros and the value of non-zeros separately [20,21,22], and it would be difficult to integrate these two different sets of coefficients together to reflect the strength of communication between cell types. To that end, we utilize the compound Poisson–Gamma distribution to model the communications scores $C_{i,j}^{k}$. This distribution effectively models the zero-inflated continuous values observed in the data while simplifying the modeling process and enhancing interpretability.

### 2.3. Compound Poisson–Gamma Distribution

For the compound Poisson–Gamma distribution CPG(λ,α,γ), the random variable *C* can be generated as follows:(3)$$C=\sum_{i=1}^{T}X_{i},\;\;T∼\mathrm{Poisson}\left(\lambda \right),\;\;X_{i}\overset{iid}{∼}\mathrm{Gamma}\left(\alpha ,\gamma \right),\;\;T⊥\;\;\;⊥X_{i},$$
where λ is the rate of the Poisson distribution, and α and γ are the shape and scale of the Gamma distribution, respectively. Based on the settings in Equation (Equation 3), we have
(C|T=t)=0ift=0,
(C|T=t)∼Gamma(tα,γ)ift>0,
which implies that the joint distribution of *C* and *T* is
(4)$$\begin{array}{cc} p(c,t|\lambda ,\alpha ,\gamma ) & =p(c|t,\alpha ,\gamma )p(t|\lambda )\\ \; & =\left\{\begin{array}{cc} \exp(-\lambda ), & \mathrm{if}\;t=0,\\ \frac{c^{\alpha t-1}}{\gamma^{t\alpha }\Gamma \left(t\alpha \right)}\exp\left(-\frac{c}{\gamma }\right)\times \frac{\lambda^{t}}{t!}\exp\left(-\lambda \right), & \mathrm{if}\;t>0. \end{array}\right. \end{array}$$

Usually, one would integrate out *T* from Equation (Equation 4) to obtain a marginal distribution of *C*. However, the infinite summand $p\left(c|\lambda ,\alpha ,\gamma \right)=\sum_{t=0}^{\infty }p\left(c,t|\lambda ,\alpha ,\gamma \right)$ does not have a closed-form representation. We can only use approximation approaches, such as series expansion [23] or Fourier inversion [24], to approximate the infinite number of terms. Although several studies have conducted statistical estimation and inference for the marginal distribution of *C* based on the approximation [25,26], in this paper, our methodology uses the joint distribution of *C* and *T*. Our approach is related to the EM algorithm presented in [27], although we use a Bayesian data augmentation strategy.

The compound Poisson–Gamma distribution CPG(λ,α,γ) is equivalently known as the Tweedie distribution TW(μ,ϕ,p) when 1<p<2. The Tweedie parametrization gradually shifts from a Poisson distribution to a Gamma distribution as *p* increases. Building a compound Poisson–Gamma generalized linear model in terms of the parameters of TW(μ,ϕ,p) is easier than the original CPG(λ,α,γ). Thus, it is critical to know the unique relationship between two sets of parameters (μ,ϕ,p) and (λ,α,γ) as follows:(5)$$\left\{\begin{array}{c} \lambda =\frac{\mu^{2-p}}{\phi (2-p)},\\ \alpha =\frac{2-p}{p-1},\\ \gamma =\phi \left(p-1\right)\mu^{p-1}, \end{array}\right.\;\;\Leftrightarrow \;\;\left\{\begin{array}{c} \mu =\lambda \alpha \gamma ,\\ \phi =\frac{\lambda^{1-p}{\left(\alpha \gamma \right)}^{2-p}}{2-p},\\ p=\frac{2+\alpha }{1+\alpha }. \end{array}\right.$$ Smyth [28] provides a detailed description of the computational process.

Returning to the communication scores $C_{i,j}^{k}$, we will utilize the Tweedie parameterization for model specification. The mean $\mu_{i,j}^{k}$ is parameterized through Equation (Equation 2) using a log-link function for g(); hence, the parameters of interest $\beta_{g_{1},g_{2}}$ determine this mean of $C_{i,j}^{k}$. The other parameters ϕ and *p* are global, and their values are shared across all (i,j) pairs of spots.

### 2.4. Model Inference

#### 2.4.1. Parameter Estimation

Considering the complexity and intractability of the proposed model, a Bayesian approach for inference will be a good choice. We use the Hamiltonian Monte Carlo (HMC) algorithm to make the sampling more efficient than the usual Gibbs sampling [29]. Because HMC requires the gradient of the log-posterior density function, we derived the closed-form solutions for it in Appendix A. The closed-form solutions are computationally less intensive and hence much faster than iterative methods.

As part of the Bayesian model specification, prior distributions for all parameters must be specified. Given the limited information available on the parameters, it is often preferable to choose weakly informative priors that offer both convenience and simplicity. Thus, the intercept term will have a disperse $N(0,100^{2})$ prior, and the remaining coefficients $\beta_{g_{1},g_{2}}$ have N(0,1) priors. As for the other two Tweedie parameters ϕ(ϕ>0) and p(1<p<2), we transform them to logϕ and $\theta =\log\left(\frac{p-1}{2-p}\right)$ and assign a normal distribution and a logistic distribution as priors, respectively. Overall, the priors are
(6)$$\begin{array}{cc} \beta_{0}\; & ∼\;N(0,100^{2}),\\ \beta_{g_{1},g_{2}}\; & \overset{iid}{∼}\;N\left(0,1\right),g_{1}=1,2,\ldots ,G,g_{2}=1,2,\ldots ,G,\\ \log\phi \; & ∼\;N(0,10^{2}),\\ \theta =\log\left(\frac{p-1}{2-p}\right)\; & ∼\;\mathrm{Logistic}(0,1). \end{array}$$

In addition to the parameters specified above, our methodology relies on the spatial tuning parameter ρ in Equation (Equation 2). This parameter is unknown; however, we can fit multiple models with varying values of ρ and select the most suitable model by considering model selection statistics.

Equation (Equation 2) reflects a mixed-effect model that accounts for both fixed and random effects. The random effects $\nu_{i}^{L}$ and $\nu_{j}^{R}$ are assumed to be independently and identically distributed according to the standard normal distribution N(0,1). In an initialization step, we fit this mixed-effects model using a Newton–Raphson algorithm on the posterior distribution. However, since the random effects are not our primary parameters of interest, we treat the estimated $\nu^{L}$ and $\nu^{R}$ as fixed parameters in the main Bayesian framework to reduce the computational burden.

As noted previously, we are using the joint distribution of (C,T) as the relevant likelihood function in our Bayesian model specification since it has a closed-form representation. Thus, every observation $C_{i,j}$ has a corresponding unknown latent variable $T_{i,j}$, and our HMC algorithm includes a data augmentation Gibbs step to sample values of *T* given the observed *c* and the current parameter values. We note that when C=0, *T* must be equal to zero. *T*s for the non-zero *C*s are sampled from probabilities proportion to $p_{\mathrm{Poisson}}\left(t^{\prime }|\lambda \right)\times p_{\mathrm{Gamma}}\left(c|t^{\prime }\alpha ,\gamma \right)$ for $t^{\prime }=1,2,\ldots ,T_{\max}$. $T_{\max}$ is an adaptive parameter in our algorithm that is increased and decreased depending on how large the sampled *T*s are relative to this maximum threshold.

In general, our Markov chain Monte Carlo (MCMC) sampling algorithm consists of two steps. In one step, we apply HMC to jointly update the vector of model parameters $\left(\beta_{0},\beta_{g1,g2},\phi ,p\right)$ for the current values of *T*. The other step updates the augmentation parameters *T* for the non-zero communications, given the current parameter values. We note that there are often few changes to the *T* values, so it is more computationally efficient to run multiple steps of HMC for each update to the augmentation parameters. Typically, we consider 10 HMC steps for every single update of *T*.

During the implementation of the MCMC procedure, we first conduct a preliminary run of 6500 iterations to tune the maximum threshold $T_{\max}$ and the covariance matrix of the momentum variables of HMC, so that the acceptance rate of all parameters can be kept around 45% to 65%. We then run an additional 13,500 iterations and discard the first 3500 iterations as the burn-in period, at which point approximate convergence is achieved. This results in 10,000 retained samples, and the inference is made using this collection.

#### 2.4.2. Hypothesis Testing

As previously mentioned, the regression coefficients $\beta_{g_{1},g_{2}}$ from Equation (Equation 1) are the main parameters of interest because they reflect the association between the cell-type communication and the LR interaction. Thus, for each coefficient, we test the null hypothesis $H_{0}:\beta_{g_{1},g_{2}}=0$ against the alternative hypothesis $H_{a}:\beta_{g_{1},g_{2}}\neq 0$. We calculate the mean and variance of the samples $\beta_{g_{1},g_{2}}$ from the inference period of HMC and then generate the statistic $W=\frac{{\hat{\beta }}_{g_{1},g_{2}}^{2}}{\mathrm{var}(\beta_{g_{1},g_{2}})}$ for a Wald test. As stated in [30], the standard Bayesian large sample theory implies that the test statistic *W* approximately follows an asymptotic $\chi^{2}\left(1\right)$ distribution under the null hypothesis, and we can easily obtain a pseudo-*p*-value by considering the tail probability beyond the *W* associated with the estimated coefficient ${\hat{\beta }}_{g_{1},g_{2}}$. The pseudo-*p*-values across all interactions of cell types will be adjusted for multiple hypothesis testing using the false discovery rate (FDR) correction [31]. In this way, the inference will be considered in a frequentist framework for ease of interpretation. In this paper, we declare a CCC significant on an LR pair if the FDR adjusted *p*-value is less than 0.05.

## 3. Results

We validated our modeling strategy through extensive simulation studies using data generated separately from two distinct models: our proposed compound Poisson–Gamma model and a pseudo-hurdle Gamma model. To accurately represent the spatial positions, we created a panel of 100 spots arranged in a 10×10 grid. Due to a lack of a comparable model structure among the existing CCC methods, we can only compare our model to variations to the structure of our BATCOM model. In the subsequent sections, we present the results of our simulation studies in Figure 2, Figure 3, Figure 4 and Figure 5. For more detailed numerical results, please refer to Appendix B. Following that, we applied the proposed model to a real dataset and compared the results with other commonly used spatial CCC methods.

### 3.1. Simulation Study

#### 3.1.1. Data Generated from the Compound Poisson-Gamma Model

To evaluate performance, we applied our method to simulated data generated from the proposed Tweedie model. Across simulations, we varied the number of cell groups *G*, the communication constraint parameter ρ, the sparsity rate of coefficients δ, and the two Tweedie parameters ϕ and *p*. For the proportions of cell types $M_{i,1},\ldots ,M_{i,G}$, we randomly generated each element from U(0,1) and then re-scaled each row so they sum to 1. Based on the regression structure, a dense vector of original coefficients $\beta_{O}$ was independently sampled from $\frac{1}{2}U\left(0.1,0.5\right)+\frac{1}{2}U\left(-0.5,-0.1\right)$, and the random effects $\nu^{L}$ and $\nu^{R}$ were sampled from N(0,0.4) independently. Considering the sparsity rates of coefficients, we randomly picked $\delta \times G^{2}$ coefficients (excluding the intercept) in $\beta_{O}$ and set them as zeros to get $\beta_{\delta }$. To make different scenarios comparable, we fixed the $\beta_{\delta }$, $\nu_{L}$, and $\nu_{R}$ across all corresponding scenarios. We also added −2 to the intercept to make the mean value $\mu =\exp(X\beta_{\delta }+\nu_{L}+\nu_{R})$ similar to real SRT data. For each combination of parameters, we generated 100 different simulated datasets to avoid uncertainty and detect variability.

For each dataset, the inference of BATCOM was performed by estimating and fixing the random effects, running the MCMC algorithm, and considering FDR-corrected Wald tests as discussed in Section 2.4. After obtaining these results, we generated a confusion matrix for each simulated dataset by comparing the results with the true values. A true positive (TP) was recorded when the estimate had the same sign as the true value and the adjusted *p*-value was less than 0.05. Similarly, a true negative (TN) was counted when the adjusted *p*-value was larger than 0.05, and the true value was zero. Conversely, a false negative (FN) was registered if the adjusted *p*-value was greater than 0.05, but the true value was not zero. Any estimate with an adjusted *p*-value less than 0.05 for a true value of zero or an estimate with a sign different from the true value was counted as a false positive (FP). Using the confusion matrix from the analysis of each generated dataset, we calculated the true positive rate (TPR), false positive rate (FPR), and observed FDR. We also plotted the receiver operating characteristic (ROC) curve by setting different cutoffs of the adjusted *p*-value to compute the area under the curve (AUC). For each scenario, we calculated the mean and standard deviations of these measurements across 100 simulated datasets.

Because we treat the communication constraint parameter ρ as a tuning parameter and do not estimate it during MCMC, it is critical to check the performance of the proposed model when the estimated value is close to and far away from the true value. Figure 2 presents the results of the proposed models fitted using $\hat{\rho }=0.2,0.5,0.8$ under the scenarios with the true values of ρ=0.2,0.4,0.6,and0.8, separately. Additionally, we applied a standard Bayesian model selection strategy to choose the best value for this tuning parameter. To that end, we consider the widely applicable information criterion (${\mathrm{WAIC}}_{2}$) [29] according to the formula
$${\mathrm{WAIC}}_{2}^{\rho }=2\sum_{i=1}^{n}\mathrm{var}\left(\log p(c_{i}|\rho ,\beta^{s},\phi^{s},p^{s})\right)-2\sum_{i=1}^{n}\log\left(\frac{1}{S}\sum_{s=1}^{S}p\left(c_{i}|\rho ,\beta^{s},\phi^{s},p^{s}\right)\right),$$
where i=1,2,…,n reflects the observation and s=1,2,…,S represents the iteration of the inference period of HMC. For each dataset, we fit the model under $\hat{\rho }=0.2,0.5,0.8$ and select the $\hat{\rho }$ that yields the lowest ${\mathrm{WAIC}}_{2}$ as the BEST model choice.

In Figure 2, the results indicate that when the $\hat{\rho }$ used to fit the model is close to the true value, the performance is excellent, with high TPR and AUC, as well as low FPR and observed FDR. We also consider the value of $\hat{\rho }=0.5$ as a useful default choice of the distance tuning parameter since it straddles the expected range of communication parameters (0,1). Empirically, this value performs well across the full range of ρ with only minor degradation in the more extreme cases of ρ=0.2,0.8. Thus, if running only one version of the model, we recommend using $\hat{\rho }=0.5$ as the tuning parameter. For the remainder of this paper, we use the default $\hat{\rho }=0.5$ choice. When computational resources permit, we suggest trying a small collection of ρ and selecting the best model based on ${\mathrm{WAIC}}_{2}$.

We then compared our Bayesian framework (BATCOM) with a frequentist framework (referred to as TWGAM) using the *gam* function from R package mgcv [32]. TWGAM is a frequentist implementation of our model structure with the same distribution assumption and design matrix (scaled by distance) but without any random effect terms. Moreover, we compared our model structure concerning the proportions of cell types in each spot with other algorithm structures considering just one cell type. Many existing CCC methods treat each spot as containing only one cell type without considering the heterogeneity in each spot. To mimic this phenomenon, we constructed a corresponding zero-one matrix. For each spot (i.e., each row of the *M* matrix), we assigned a 1 to the cell type with the maximum proportion and a 0 to the rest of the cell types (named MAXPROP). MAXPROP uses all the other features of our proposed BATCOM except that the membership proportions $M_{i,g}\;\left(i=1,\ldots ,N;g=1,\ldots ,G\right)$ are binary. For this MAXPROP, we employed our overall framework, including the random effect imputation, the MCMC sampling strategy for parameter estimation, and the Wald tests for inference. In addition, we compared our distribution assumption with the binomial logistic distribution of non-zero values (named LOGISTICS), which does not account for random effects.

Figure 3 displays the performance of different models using $\hat{\rho }=0.5$ as the tuning parameter value under different scenarios, where the true communication constraint parameter ρ is either 0.4 or 0.6. It is easy to see that our proposed model (BATCOM) maintains a very robust performance across these different scenarios, consistently achieving high TPR and AUC when compared to three other models. Remarkably, BATCOM also effectively controls the FPR, and the observed FDR was around or below 0.05. In contrast, the frequentist framework (TWGAM), which shares the same distribution assumption as BATCOM, consistently performs worse, with much higher FPR and FDR. One possible explanation for this result is that TWGAM does not integrate prior knowledge about the parameters, and the failure to account for random effects could contribute to an increased FDR. The model that only considers one cell type for a spot (MAXPROP) is even worse than TWGAM in all aspects, including FPR, FDR, and AUC, with particularly high FPR and FDR. To some extent, these results suggest that existing CCC methods that ignore within-spot heterogeneity may produce numerous false discoveries. Additionally, although the model with the Bernoulli distribution assumption (LOGISTICS) generally maintains a small FPR and FDR, it has the lowest power (TPR), especially when we have a high value of the Tweedie parameter *p*. In other words, it tends to make too-conservative conclusions by inferring that most interactions among cell types are insignificant.

In our previous simulation results, we considered all spot pairs for analysis. However, in real-world applications, we can effectively reduce the number of observations in the model by focusing only on spot pairs (i,j) that are within a specific distance threshold of each other, as mentioned earlier. To investigate the trade-off between efficiency and accuracy resulting from this reduction, we conducted an additional simulation experiment by varying the distance threshold and evaluating its impact on estimation accuracy. Specifically, we set the threshold values to 10 (i.e., including all spot pairs), 7, 5, and 3, as illustrated in Figure 4. It is evident that reducing the threshold to include fewer (i,j) spot pairs in the model inevitably affects estimation accuracy; however, the performance did not deviate significantly. The TPR and AUC showed a significant reduction when the threshold was set to 3, but the change remained below 10%. In contrast, the FPR and FDR exhibited no significant changes with varying thresholds. This observation suggests that reducing the number of spot pairs during model fitting leads to a more conservative decision-making process in our algorithm.

It is worth noting that the choice of the threshold is closely related to the selection of ρ. If ρ is large, only the closest spots have a substantial contribution, making a small threshold appropriate. However, when ρ is small, more spots contribute, and a larger threshold may be necessary to avoid excluding relevant spots. In Figure 4, we used $\hat{\rho }=0.5$, resulting in a minimal weight for the exponential term of distance in Equation (Equation 2) when the distance between two spots exceeds 5.

#### 3.1.2. Data Generated from the Pseudo-Hurdle Gamma Model

To complete our comprehensive evaluation of performance, we also simulated data from a structure that differs from our methodology while maintaining a comparable set of model parameters describing the relationship between cell type memberships and communication scores. We generated this additional simulated data from a pseudo-hurdle Gamma model. In the traditional hurdle model, one needs to have two different sets of coefficients to determine the probability of zeros and the distribution of non-zero values separately. Here, we generated the new data using the model
$$\mathrm{Pr}\left(C_{\mathrm{new}}=0\right)=1/\left(1+\mu \right),$$
$$C_{\mathrm{new}}|C_{\mathrm{new}}>0∼\mathrm{Gamma}\left(\alpha_{\mathrm{new}},\frac{1+\mu }{\alpha_{\mathrm{new}}}\right),$$
where μ depended on the same βs through Equation (Equation 2), the shape parameter $\alpha_{\mathrm{new}}$ was randomly selected from a uniform distribution U(0.5,5), and the scale parameter $\gamma_{\mathrm{new}}=\frac{1+\mu }{\alpha_{\mathrm{new}}}$. In this way, the simulated data from this pseudo-hurdle Gamma model have the same mean μ as our proposed compound Poisson-Gamma distribution, ensuring that the parameter interpretation of β is comparable.

After switching to the pseudo-hurdle Gamma model as the data generator for our simulation scheme, we found that the results remained consistent with our previous findings using the compound Poisson–Gamma distribution. As shown in Figure 5, our proposed model outperforms other models, particularly in terms of its high TPR and low observed FDR in all scenarios. Even when faced with higher sparsity rates of coefficients and a larger number of cell types, our proposed model strikes a balance between identifying new discoveries and minimizing errors.

### 3.2. Case Study

We applied our methods to the Visium spatial transcriptomics data of cutaneous squamous cell carcinoma (cSCC) [33], in which each spot contains multiple cells. Ji et al. [33] performed scRNA-seq on both tumors and normal skin and profiled SRT data on tumors simultaneously. As an example, we focused on the SRT data from replicate 2 of patient 2, which had a greater sequencing depth than other samples. This sample contains 1932 spots (after excluding spots with less than 100 genes detected) and 10,703 genes (after filtering out genes not expressed in at least 97.5% of the spots). To perform the upstream cell type deconvolution analysis, we utilized scRNA-seq data from the same patient as a reference and obtained the cell type proportion matrix using the full mode of RCTD [16].

Subsequently, we conducted a comparative analysis between our proposed method, employing $\hat{\rho }=0.5$ as the tuning parameter, and other CCC algorithms tailored to SRT data, including Giotto [6], COMMOT [8], and SpaTalk [9]. To ensure a fair comparison, we utilized the same cell type proportion matrix from RCTD [16] (G=24 cell types) and the same list of known LR pairs from CellTalkDB [19] (3398 pairs). However, Giotto [6] and COMMOT [8] required a single cell type to be specified for each spot, which posed a challenge for comparison. To address this, we assigned to each spot the cell type that had the highest proportion in the matrix, which introduced 12 cell types into the algorithms. Furthermore, we fit our model using the MAXPROP version, which is expected to be suboptimal for spot-based SRT data.

After filtering out LR pairs based on each algorithm’s default rules, we found that our BATCOM and MAXPROP methods considered 712 LR pairs, whereas SpaTalk considered 515 LR pairs, Giotto considered 983 LR pairs, and COMMOT focused on 664 LR pairs. Regarding cell-type interactions, while there were a total of 576 interactions possible among the 24 cell types, we filtered out some interactions due to minimal or non-existent observations; this resulted in 364 interactions (out of 576) for BATCOM and 96 interactions (out of 144) for MAXPROP. SpaTalk does not consider interactions between the same cell type, leading to 552 interactions under its consideration. Meanwhile, Giotto and COMMOT dealt with 144 interactions due to the presence of 12 cell types.

Figure 6 illustrates the UpSet plot for the number of significant CCC on LR pairs identified by different methods. As we can see, our proposed method, BATCOM, and MAXPROP detected the third and second highest number of significant communications, respectively, following SpaTalk, which identified the most significant results. It is not surprising that our proposed model (BATCOM) identified fewer significant CCC results than SpaTalk, as SpaTalk does not correct the *p*-values for multiple comparisons. In contrast, Giotto and COMMOT found the fewest communication pairs. Notably, MAXPROP identified many more significant CCCs than BATCOM, which is consistent with the higher FDR observed in the preceding simulations. Compared to Giotto and COMMOT, which also utilize the modal cell type, MAXPROP found more significant communications. Specifically, one-third to one-half of the results from Giotto and COMMOT overlapped with MAXPROP but disagreed with BATCOM, which is the version of our model that uses all available information about the spot’s cell type makeup.

All methods share a common finding of six significant CCC pairs (Table 1). The results suggest that several LR interactions between different cell types may play a critical role in tumor-specific cellular crosstalk. Specifically, the ligand *SERPINE1*, which binds to the receptor *ITGB5*, has been found to promote tumor growth and angiogenesis in several types of cancer, including skin cancer [34]. Similarly, *THBS1*, *SDC4*, and *TLN1* have been linked to the development of metastasis and chemoresistance in skin cancer [35,36,37]. Notably, previous research has established *PLAU* and *ITGA5* as critical biomarkers for various types of squamous cell carcinoma [38,39,40,41]. Furthermore, the study by Fang et al. [39] suggests that *PLAU* affects the formation of inflammatory cancer-associated fibroblasts, which is consistent with the findings of our CCC analysis. These results emphasize the crucial role of specific LR interactions in cancer progression and highlight potential targets for therapeutic interventions.

In addition to inferring CCC for a specific LR pair, exploring the overall cellular communication or communication within a specific signaling pathway based on SRT data can provide valuable insights. To demonstrate this, we aggregated the communication scores across all LR pairs and fit BATCOM using $C_{i,j}=\sum_{k}C_{i,j}^{k}$ as the outcome variable in Equation (Equation 2).

Figure 7 depicts the overall significant CCCs, indicating that cancer-related cells communicate closely with each other. Specifically, normal-differentiating keratinocytes (KC) exhibit positive communication with tumor-differentiating KC and plasmacytoid dendritic cells (PDC) while showing negative communication with fibroblasts. Notably, Ji et al. [33] found that the subpopulations of tumor KCs (basal, cycling, and differentiating) closely resemble the normal KC subpopulations, and they identified a fourth major tumor KC subpopulation, called tumor-specific keratinocytes (TSK), that exclusively exists in tumor skin and distinguishes itself from other tumor cells. Furthermore, Ji et al. [33] discovered that TSK and tumor-basal KC are both present in the leading edge of the tumor. Our results are consistent with these findings, as Figure 7 shows that tumor-basal KC frequently communicates with TSK.

## 4. Discussion

In this paper, we present a generalized linear regression model for inferring CCC based on LR interactions. Our model offers high flexibility in fitting both the single-cell resolution SRT data and spot-based SRT data. A significant challenge in spot-based SRT data is the presence of cell type mixtures in each spot, which we address by assuming that the mean spot-to-spot communication score is a convolution of possible interactions between cell types at the sender spot and the receiver spot. Our proposed model takes advantage of the regression model’s properties to naturally handle communication between different cell types simultaneously, while also directly providing the direction of the association between CCC and LR interaction. Furthermore, our approach explicitly models the decreasing ability to communicate as the distance between cells or spots increases, differing from other algorithms that employ an arbitrary threshold to restrict communication.

Due to the limited information available on the parameters, our detailed Bayesian algorithm assumes the prior distributions defined in Equation (Equation 6). It is crucial to recognize that the choice of different prior distributions can lead to diverse model performances. To explore this, we also examined alternative priors such as $N(0,0.1^{2})$ and $N(0,0.01^{2})$ for the regression coefficients $\beta_{g_{1},g_{2}}$, and we observed that the model’s inference regarding significant connections remained robust relative to using the default N(0,1) prior. However, if necessary, the prior standard deviation can be easily adjusted in practice.

When comparing Bayesian and frequentist inference with the same distributional assumptions, we found that Bayesian inference provides more accurate estimation with lower FDR (Figure 3). However, MCMC algorithms can be time-consuming. For instance, in the cSCC case study, BATCOM and MAXPROP had an average running time of 67.38 and 19.80 min, respectively, per LR pair. In contrast, COMMOT and Giotto had average running times of 2.16 and 0.18 min, respectively, for each LR pair. SpaTalk had a different approach, identifying significant LR pairs for each cell type interaction, with an average running time of 4.38 min per interaction. To reduce the computational burden, one potential solution is to employ a threshold to control the number of included (i,j) spot or cell pairs. Although this approach inevitably impacts estimation accuracy, our simulation results (Figure 4) demonstrate that the resulting influence on performance will be minimal if using a moderate or large $\hat{\rho }$, such as 0.5. Alternatively, further exploration could focus on developing a more precise frequentist inference framework for the compound Poisson-Gamma distribution.

Currently, in this paper, we have defined the communication score as a product of the arithmetic mean of the ligand expressions at the sender spot and the mean receptor expression of the receiver spot. This simple approach implicitly assumes equal importance across the subunits of the LR pair, which may not always be the case. Certain subunits could have varying weights or specific distributions, necessitating a more sophisticated strategy in the future to accurately account for their expression. Moreover, we only considered the simple multiplication of ligands and receptors as the communication score between cells/spots (Equation (Equation 1)), while some other algorithms, such as CellChat [5], consider more complex relationships between ligands and receptors, including agonists and antagonists. It is certainly possible to design more intricate communication scores between cells/spots by accounting for these relationships. Given the versatility of our model, we can apply our approach directly to communication scores that have numerous zeros and positive continuous data, regardless of their complexity. This flexibility enables us to adapt our method to various scenarios and extend its applicability in future studies.

As previously mentioned, the tuning parameter ρ in Equation (Equation 2) is responsible for controlling the rate of decay of spot-to-spot communication as the distance between two spots increases. The appropriate value of ρ depends on the distance unit and potential communication assumption used in a specific tissue. While the parameter is not estimated during MCMC, we recommend using $\hat{\rho }=0.5$ as a default value based on our simulation study results (Figure 2). For those with more computational resources, we suggest experimenting with different values of ρ and selecting the best one based on ${\mathrm{WAIC}}_{2}$. For this manuscript, the utilization of Euclidean distance in our proposed model to account for the spatial proximity of cells or spots is specifically due to the current SRT data being derived from tissue slices relying on Cartesian coordinates. If future advancements in SRT technology enable the measurement of tissue shapes beyond the current capabilities, it will become imperative to explore alternative distance measurements that are better suited for such scenarios.

Furthermore, in the proposed model, we assume a decreasing trend in the communication probability when the distance between cells/spots increases. However, long-distance signaling is also essential in biological activities [42]. Therefore, a more comprehensive consideration of the relationship between communication and distance should be a focus of future research.

The results presented in Figure 6 indicate that various algorithms yield highly divergent results, with each method exhibiting a substantial number of distinct significant CCC. This observation aligns with the prior work by Li et al. [43]. It should be noted that the inferior performance of Giotto and COMMOT in our study may be partially attributed to directly assigning the cell type with the highest proportion to each spot. However, it is important to highlight that these CCC algorithms only allow for one cell type per spot. To ensure a fair comparison, we implemented the MAXPROP version of our model structure, aligning with the basic design of these algorithms. While we are confident that our methodology is statistically rigorous and reliable, the significant disparities across the different methods make it difficult to determine the most appropriate method at the biological level. Adding to this challenge is the lack of ground truth in this research domain. Moreover, although we conducted simulation studies with two different distributions to evaluate our model’s performance, these assessments still rely on the underlying structure of our algorithm. Thus, it is imperative to undertake further experimental investigations and validations of CCC analysis to determine the most appropriate method for this area of inquiry.

As the field of CCC analysis continues to grow with the generation of more SRT data, we believe that our proposed model will serve as a valuable approach for inferring cellular communication in a flexible and accurate way. Our innovative approach can bridge gaps in current CCC inference methods and provide a straightforward outcome. Future studies could explore the applicability of our model to other more complicated structures and further validate its effectiveness in real-world scenarios.

## Figures and Tables

**Figure 1 genes-14-01368-f001:**
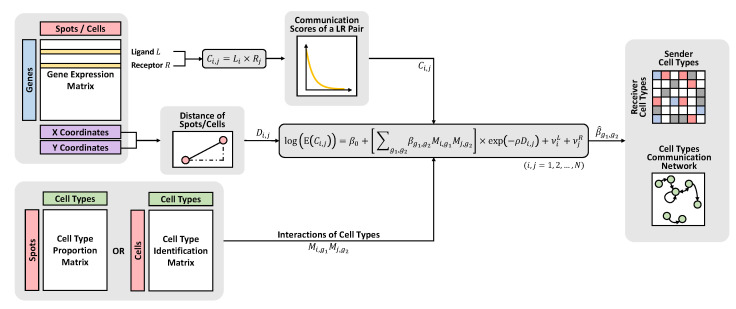
Overview of the workflow. BATCOM requires three main inputs: (1) a gene expression matrix obtained from the SRT data with coordinates information, (2) a matrix that reflects the cell type annotations of spots/cells, and (3) a list of known LR pairs. After fitting the model, a heatmap and a network can be generated for visualization purposes.

**Figure 2 genes-14-01368-f002:**
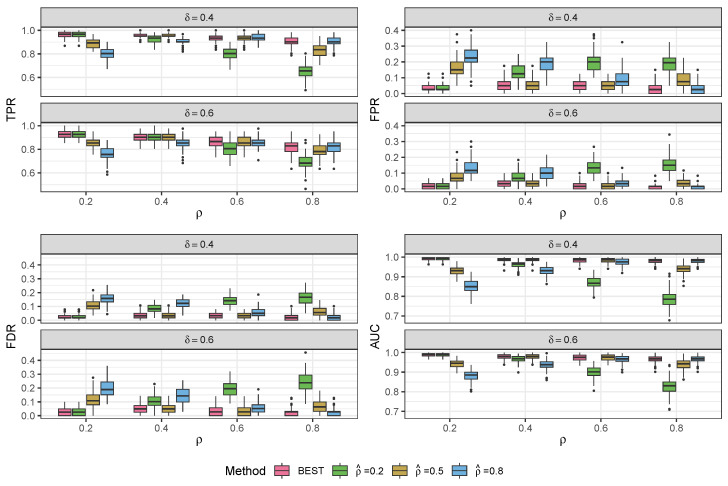
Results of BATCOM with different estimated ρ’s based on the simulation data generated from the proposed compound Poisson–Gamma model. All scenarios were G=10, ϕ=3, and p=1.5. TPR: true positive rate; FPR: false positive rate; FDR: false discovery rate; AUC: area under the ROC curve.

**Figure 3 genes-14-01368-f003:**
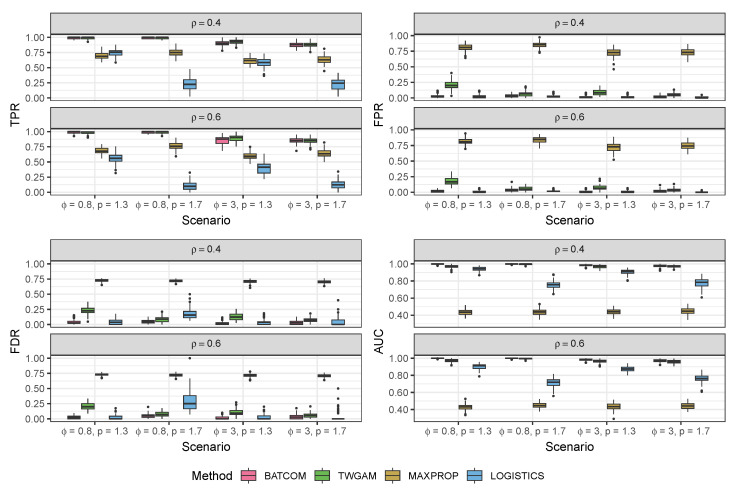
Results of different models based on the simulation data generated from the proposed compound Poisson–Gamma model. All scenarios were G=10 and δ=0.6. All methods here used $\hat{\rho }=0.5$. TPR: true positive rate; FPR: false positive rate; FDR: false discovery rate; AUC: area under the ROC curve.

**Figure 4 genes-14-01368-f004:**
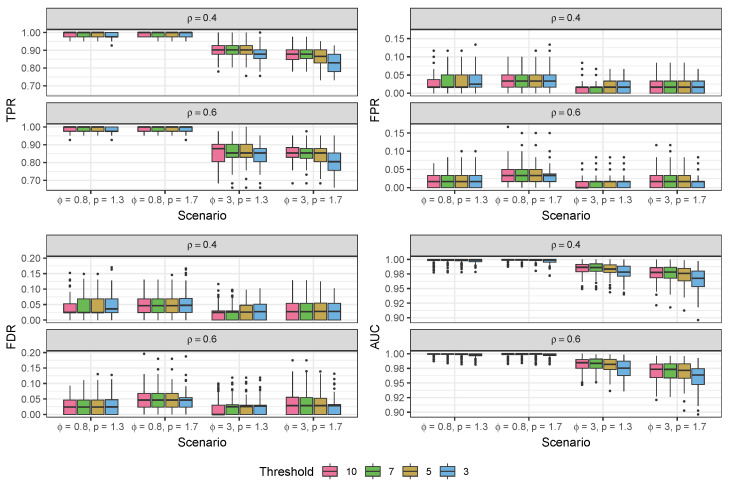
Results of BATCOM using different thresholds of distances of spots on the simulation data generated from the proposed compound Poisson–Gamma model. All scenarios were G=10 and δ=0.6. All methods here used $\hat{\rho }=0.5$. TPR: true positive rate; FPR: false positive rate; FDR: false discovery rate; AUC: area under the ROC curve.

**Figure 5 genes-14-01368-f005:**
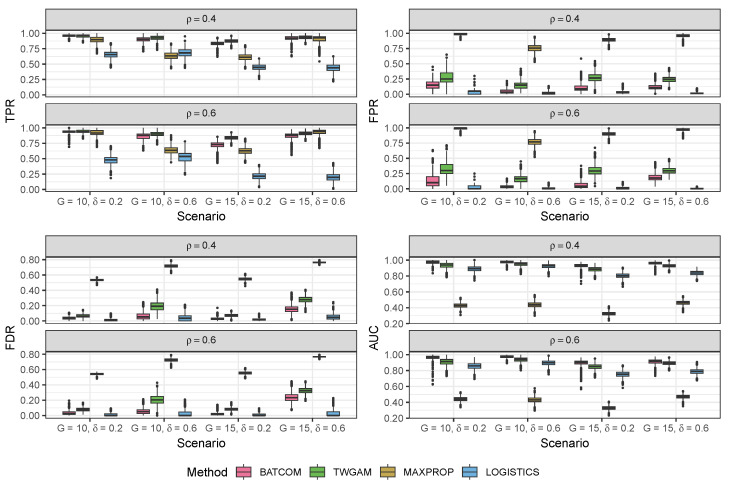
Results of different models based on the simulation data generated from the pseudo-hurdle Gamma model. All methods here used $\hat{\rho }=0.5$. TPR: true positive rate; FPR: false positive rate; FDR: false discovery rate; AUC: area under the ROC curve.

**Figure 6 genes-14-01368-f006:**
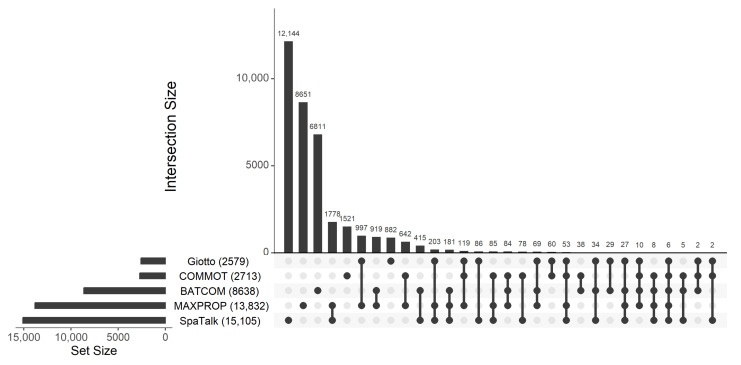
UpSet plot of significant CCC on LR pairs determined by different methods for the cutaneous squamous cell carcinoma data.

**Figure 7 genes-14-01368-f007:**
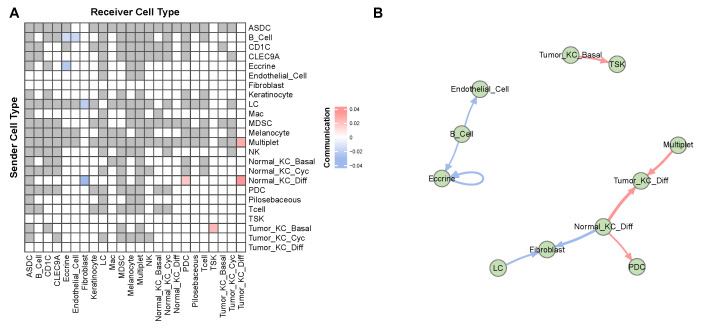
Overall CCC in the cutaneous squamous cell carcinoma data estimated by BATCOM. (**A**) Heatmap of CCC between the sender cell types and the receiver cell type. The gray blocks are the interactions of cell types that have been filtered out before fitting the model. The white blocks represent insignificant CCC. The colored blocks represent the significant CCC. (**B**) Network of CCC. The edge width reflects the strength of communication. The edge color shows the direction of the association.

**Table 1 genes-14-01368-t001:** Significant CCC pairs shared in all methods.

Ligand	Receptor	Sender Cell Type	Receiver Cell Type
*SERPINE1*	*ITGB5*	Fibroblast	TSK
*SERPINE1*	*ITGB5*	TSK	Fibroblast
*THBS1*	*SDC4*	Fibroblast	TSK
*PLAU*	*ITGB5*	TSK	Fibroblast
*TLN1*	*ITGB5*	TSK	Tumor KC Diff
*PLAU*	*MRC2*	TSK	Fibroblast

TSK: tumor-specific keratinocytes; Tumor KC Diff: tumor-differentiating keratinocyte.

## Data Availability

The cutaneous squamous cell carcinoma data analyzed in this study are openly available in the NCBI GEO database repository (https://www.ncbi.nlm.nih.gov/geo/) under accession number GSE144240 (accessed on 3 October 2022). The code scripts utilized in this study can be accessed on https://github.com/dongyuanwu/BATCOM.

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
