# Peer review of "Inferring Cell–Cell Communications from Spatially Resolved Transcriptomics Data Using a Bayesian Tweedie Model"

_genes, 2023, doi:10.3390/genes14071368_

Round 1

Reviewer 1 Report

The authors introduce BATCOM, a generalized linear regression model for inferring cell-cell communications (CCC) in scRNAseq or spatial transcriptomics. Compared to other methods tested in this work, BATCOM has two main advantages: (1) It is built on a Bayesian framework that allows for the use of spatial transcriptomic data, (2) Allowing to infer the direction of the communication, which provides additional insights. The statistical model is intuitive, however, the motivation of results and ablation studies would benefit from greater detail.

Below are some major and minor comments that will improve the manuscript and presentation of results.

Major Comments/Questions:

1) One of the novelties of BATCOM is incorporating distance in the modeling. However, can the authors describe the reliability of Euclidean coordinates of a tissue? From both the mathematical and biological perspectives, it seems like this would be highly dependent on the tissue of interest.  For example, assume a spherical organoid, which can be laid in 2D (for spatial sequencing) as a flat circle, or cut open to resemble a rectangle. Can the authors comment on the difference and address if/how the  x-y coordinate may not be informative. 

2) An issue of concern is the number of potential pairs used, which could affect efficiency and accuracy in the future. Moreover, the generalized regression model includes a “density" term (exponential of the distance matrix). The authors mention that a specific ε-ball can be defined for restricting the number of pairs. Though this makes sense, an ablation study of various different epsilon choices for accuracy/efficiency can improve the current manuscript. In addition, just as in my first point, it would be helpful for the authors to provide additional detail on the limitation of their approach/methodology.

3) Could the authors provide the reasoning behind choosing the initial parameters (e.g. β N (0, 100))? (I did not see this discussion and suspect there's a clear intuitive explanation.)

4) I am confused by the value that vL and vR add to the modeling individually. It seems the effect could be captured with just one variable, especially since they are fixed during inference. 

5) Generally, I think the presentation of the results requires some additional consideration. I believe visualizing the results (e.g. of the simulation ablation study for δ ) in the main manuscript and moving the long (and wide) tables to the appendix would improve the readability of the manuscript significantly.

Plotting the critical results in a figure would allow the readers to get a sense of what the authors are trying to show quickly. Readers interested in the fine-grained results can refer to the tables in the appendix. 

6) For the results of the Case Study, is there a better way of assigning cell types to spots for Giotto and COMMOT? Currently, the authors use cells with the highest proportion as the cell type of the spot, which reduces the cell types from 24 to 12. This may be the reason for the poorer performance.

For Giotto, the authors (Dries et al) write: "To overcome the challenge of lower resolution, Giotto implements a number of algorithms for estimating the enrichment of a cell type in different regions (Fig. 3a). In this approach, a continuous value representing the likelihood of the presence of a cell type of interest is assigned to a spatial location which contains multiple cells. To this end, Giotto requires additional input representing the gene signatures of known cell types. Currently, the input gene signatures for the known cell types can either be provided by the user directly as cell type marker gene lists, or be automatically inferred by Giotto based on an additional scRNAseq data matrix input."

Dries et al. show that their approach for deconvolution can identify cell types "even if a spot contains only one cell from a given type". Given that the scRNAseq was readily available, could this have been added to the Giotto object prior to performing CCC? 

Minor Comments:

1) I appreciate the authors providing R scripts for reproducibility (though I have not run these scripts yet). A minor comment would be adding inline comments to the scripts to improve readability.

2) The authors’ introduction to Giotto, spaOTsc, and SpaTalk is short but sufficient in the introduction section. I would recommend adding references to a few review papers on these methods (see [1] & [2]) so that the reader can look at these methods (and others similar to them) in greater detail in one place.

[1] Axel A. Almet, Zixuan Cang, Suoqin Jin, Qing Nie, The landscape of cell–cell communication through single-cell transcriptomics, Current Opinion in Systems Biology, Volume 26, 2021, Pages 12-23.

[2] A. Ali Heydari, Suzanne S. Sindi; Deep learning in spatial transcriptomics: Learning from the next next-generation sequencing. Biophysics Rev. 1 March 2023; 4 (1): 011306. 

3) Mentioning "deconvolution" explicitly may be helpful to readers when discussing using RCTD to obtain cell type proportion.

4) Line s23: "... we need their gradient" should be changed to "... we need their gradients." 

Reviewer 2 Report

Major points:

-          The authors state that “Although the geometric mean of subunits can be an alternative option for estimation, this will introduce a significantly larger number of zeros compared to the arithmetic mean.” I agree but, on the other hand, a sum of terms does not take into account the nature of the complex (all parts should be available…). There are maybe other strategies that can account for the distribution of the expression of the subunits on the spot/type of cell under consideration

-          “Thus, Equation (2) includes two random effect parameters νLi and νRj to introduce correlation around the corresponding spots for ligand Lki (i = 1, 2, ..., N) and receptor Rkj (j = 1, 2, ..., N), respectively.” To me it is not completely clear how this is achieved. Please explain.

-          The authors use a compound Poisson-Gamma distribution CPG(λ, α, γ), to model cell communication. I think that the statistical and biological/technical rationale behind this choice should be explained.

-          To evaluate performance, the authors use simulated data generated from the proposed Tweedie model. I am aware there are no golden standards in the field, but of course in this way they introduce a bias with a clear advantage for their method. They should use at least another type of simulation and state the limitation of their assessment.

Minor points:

-         I find Figure 1 not completely effective. The most original part of the method is the model (right hand side of the figure) and should take more space and more details

Round 2

Reviewer 2 Report

The authors have properly answered to all my comments